# Cocaine affects foraging behaviour and biogenic amine modulated behavioural reflexes in honey bees

Eirik Søvik[1,2], Naïla Even[1], Catherine W. Radford[1] and Andrew B. Barron[1]

[1] Department of Biological Sciences, Macquarie University, Sydney, Australia
[2] Department of Biology, Washington University in St. Louis, St. Louis, USA

## ABSTRACT

In humans and other mammals, drugs of abuse alter the function of biogenic amine pathways in the brain leading to the subjective experience of reward and euphoria. Biogenic amine pathways are involved in reward processing across diverse animal phyla, however whether cocaine acts on these neurochemical pathways to cause similar rewarding behavioural effects in animal phyla other than mammals is unclear. Previously, it has been shown that bees are more likely to dance (a signal of perceived reward) when returning from a sucrose feeder after cocaine treatment. Here we examined more broadly whether cocaine altered reward-related behaviour, and biogenic amine modulated behavioural responses in bees. Bees developed a preference for locations at which they received cocaine, and when foraging at low quality sucrose feeders increase their foraging rate in response to cocaine treatment. Cocaine also increased reflexive proboscis extension to sucrose, and sting extension to electric shock. Both of these simple reflexes are modulated by biogenic amines. This shows that systemic cocaine treatment alters behavioural responses that are modulated by biogenic amines in insects. Since insect reward responses involve both octopamine and dopamine signalling, we conclude that cocaine treatment altered diverse reward-related aspects of behaviour in bees. We discuss the implications of these results for understanding the ecology of cocaine as a plant defence compound. Our findings further validate the honey bee as a model system for understanding the behavioural impacts of cocaine, and potentially other drugs of abuse.

Corresponding author
Eirik Søvik, eirik.sovik@gmail.com

## INTRODUCTION

Humans and mammals consume drugs of abuse because they make them feel good (*Siegel, 2005*). This presents an unusual paradox (*Sullivan, Hagen & Hammerstein, 2008*), since many of the drugs of abuse are naturally occurring plant-derived compounds, and the evolutionary explanation given for the existence of most plant-derived drugs of abuse, is that they evolved as a defence mechanism to deter herbivory (*Sullivan, Hagen & Hammerstein, 2008*). It therefore makes no sense that these compounds should be consumed for their

rewarding properties and may even be consumed compulsively. An explanation given for this apparent paradox is that plants evolved to deter herbivorous insects (*Nathanson et al., 1993*), not mammals. This argument assumes that the neurochemical pathways affected by drugs of abuse do different things in these two animal groups such that drugs of abuse are lethal to insects, but rewarding to mammals. By this argument drug reward is viewed as an evolutionary side-effect as mammals are not seen as the co-evolved target of these plant defence compounds. If this explanation is correct, drugs of abuse should not be rewarding to insects.

For a while there was some support for the idea that the neurochemical pathways signalling reward and aversion differed between insects and mammals, however this view is now being revised (*Waddell, 2013*). The predominant belief was that dopamine, which signals reward in mammalian nervous systems (*Schultz, 2007*), signalled aversive stimuli in insects (*Schwaerzel et al., 2003*; *Vergoz et al., 2007*; *Honjo & Furukubo-Tokunaga, 2009*; *Nakatani et al., 2009*). However, as more precise genetic tools have become available for studying reward circuitry in insects, it has become clear dopamine plays a role in reward signalling in insects as well (*Waddell, 2013*).

Despite the similarity in neurochemical reward pathways, very few studies have examined the possibility of drug reward in insects (*Søvik & Barron, 2013*). The most convincing evidence that a psychostimulant drug can affect the reward system of an insects comes from the finding that following treatment with cocaine, bees were more likely to do a recruitment dance that is highly correlated with perceived reward value of a foraging site (*Barron et al., 2009*). This indicated that cocaine affected the perceived value of the floral resources collected.

Consequently, we investigated the effects of cocaine on reward related behaviours in honey bees. We examined whether honey bees developed a preference for a location in which they had been treated with cocaine, and whether cocaine altered foraging activity. Further, we explored the effects of cocaine on a simple appetitive reflex, sucrose responsiveness (*Scheiner, Page Jr & Erber, 2001*; *Scheiner, Page Jr & Erber, 2004*). Lastly, to test if the behavioural effects were limited to reward related behaviours we examined the effects of cocaine on responsiveness to punishing electric shock using the sting extension reflex (*Roussel et al., 2009*; *Giray et al., 2014*; *Tedjakumala, Aimable & Giurfa, 2014*). We discuss our findings in terms of understanding the actions of cocaine on insects and the implications of this for reconciling the ecological and neurobiological roles of cocaine.

## MATERIALS & METHODS

### Subjects

All experiments were performed at Macquarie University, Sydney, Australia. Bees used were of the standard commercially available strains in Australia, and reared according to standard bee keeping practices. For foraging experiments, a colony containing approximately 5,000 bees was housed in a 400 m$^2$ flight enclosure.

## Pharmacological treatments

For topical application, 3 µg freebase cocaine dissolved in 1 µL dimethylformamide (DMF) was applied to the dorsal thorax of bees using a glass microcapillary. This was the same non-toxic dose that increased dance rate in the study by *Barron et al. (2009)*. DMF is a solvent that can penetrate bees' cuticle and allows cocaine to pass into the haemocoel (*Barron et al., 2007*). This method has previously been used for administering cocaine to honey bees (*Barron et al., 2009*; *Søvik, Cornish & Barron, 2013*). As a control, bees were treated with DMF alone in the same manner.

For volatilised treatments, freebase cocaine was dissolved in ethanol, and carefully pipetted onto a nichrome wire filament connected to a power source (*McClung & Hirsh, 1998*). Ethanol was evaporated from the filament at room temperature. To treat bees, a single bee was kept in a 50 cm³ airtight container encapsulating the filament. The filament was heated for 10 s and bees were kept in the container, exposed to volatilised cocaine, for one minute. Unlike vertebrates, insects have an open gas exchange system that transports oxygen directly to tissues where it is needed in the gaseous phase, bypassing the haemolymph. Air is taken in through spiracles in the thorax and abdomen, passed through trachea, before gas exchange takes place via tracheoles (*Chapman, 2013*). This system allows volatilised cocaine to be delivered directly to cells throughout the bee nervous system. As a control, pure ethanol was applied to the filament, allowed to evaporate, and the clean filament was used for treatments using the method outlined above (for details see *Søvik, Cornish & Barron, 2013*). All reagents were supplied by Sigma-Aldrich (St. Louis, MO, USA).

Previously we have shown that the pharmacokinetics of these two methods are markedly different (*Barron et al., 2009*; *Søvik, Cornish & Barron, 2013*), but without measuring the rate cocaine enters and is cleared from the brain following administration it is not possibly to conclusively state how different.

## Effects of cocaine on honey bee foraging preferences

To examine if bees developed a preference for a feeder associated with cocaine treatment, 60 individually paint-marked bees were trained to two *ad libitum* 1.5 M sucrose feeders placed at the closed ends of two 2 m long tunnels that intersected at a 45° angle (Fig. 1). The walls and floor of the tunnels were solid opaque plastic; the ceiling was covered with mesh. From the perspective of approaching from the hive, the entrance to the left tunnel and the walls surrounding the feeder in the left tunnel were marked with horizontal green and white stripes, the entrance to the right tunnel and the walls surrounding the feeder in the right tunnel were marked with vertical blue and white stripes. The tunnels created two visually distinct and spatially separated environments in which feeders were located. The bee's choice of feeder could easily be assayed visually by observing which tunnel they entered and which feeder they alighted on. The colours blue and green were chosen because bees have distinct photoreceptors for these two colours (*Chittka & Menzel, 1992*), further, the 90° difference in orientation of the striped patterns is easily differentiated by honey bees (*Frisch, 1971*) and was added to make the tunnels even more distinctive.

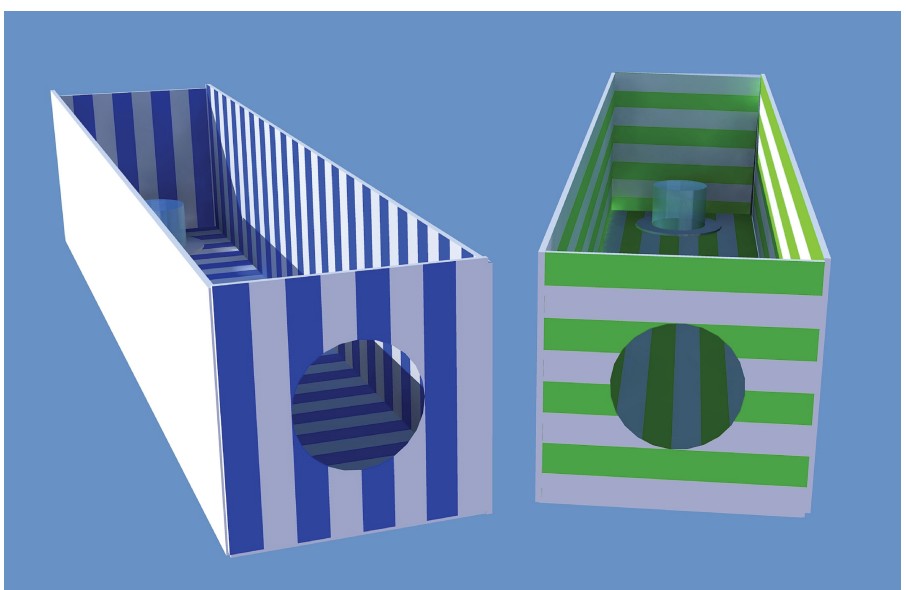

**Figure 1 Schematic of experimental set-up used for foraging preference experiment.** In the foraging preference experiment, bees were trained to two tunnels. One was blue with vertical stripes while the other was green with horizontal stripes. The difference between the two tunnels was to make it as easy as possible for the bees to tell the two tunnels apart.

This design was chosen in order to increase the distinctiveness of the two tunnels (i.e., in order to make it as easy as possible for the bees to tell the two tunnels apart). This allowed detecting changes in preference rather than discriminatory abilities.

Bees were trained and tested in a five-day protocol. On day one of a trial, bees were trained to use both tunnels by alternating the availability of tunnels every 15 min while progressively stepping a 1.5 M sucrose feeder deeper into each tunnel over a 4 h period. Bees were released from the tunnel after feeding by lifting the mesh.

On day two, bees were further trained to use the tunnels by alternating the availability of the tunnels every 30 min for 3 h, then simultaneously opening both tunnels to provide bees with a free choice of feeders for 1 h. During this time the number of visits of each bee to each feeder was recorded. These were converted to a preference index as follows:

$$\text{Preference index} = \frac{(\text{number of visits to green tunnel} - \text{number of visits to blue tunnel})}{\text{total number of visits}}$$

This preference index is similar to that used for aversive conditioning by *Vergoz et al. (2007)*, but because individual bees varied in the total number of visits made, we divided difference in visits made by the total number of visits to allow for comparison between bees. At this stage the median preference index was not significantly different from zero (Wilcoxon signed rank test. $W = 456$, $p = 0.166$, $n = 75$) indicating there was no preference toward either tunnel.

On days three and four of a trial, bees had access to the green tunnel only for 2 h a day, which offered a 1 M sucrose feeder. Bees were randomly assigned to cocaine or control

treatment groups. We used the slower topical treatment method so that cocaine would persist in bees' systems for the majority of their time interacting with the tunnels (previous work suggested topical cocaine treatment influenced bee behaviour for approximately 1.5 h following treatment (*Barron et al., 2009*), whereas the effects of volatilised treatment appeared to be shorter in duration). Bees were treated with either 1 μl DMF containing 3 μg cocaine or 1 μl DMF alone on their first visit to the feeder each day.

With this assay design bees had more opportunities to visit the green tunnel than the blue tunnel, and therefore had more reinforcing experiences in the green tunnel than the blue tunnel. Thus, we expected all bees to develop a weak preference for the green tunnel. However, the aim of this experiment was to test whether cocaine treatment affected the magnitude of the preference for the green feeder.

On day five of a trial, all bees were given simultaneous access to both tunnels for 1 h to test the preference of bees for the different tunnels. The number of visits by each bee to each tunnel was recorded. During the test both tunnels contained empty feeders, and once bees had reached the end of a tunnel they were released. The number of visits to each tunnel by each bee was converted to a preference index as described. Five replicate trials of this experiment were performed. For analysis data from all trials were pooled.

## Effect of volatilised cocaine on foraging rate

Previously, *Barron et al. (2009)* did not find a difference in foraging rate between bees treated with cocaine and controls, using the topical treatment method. As topical treatment is rather slow (*Barron et al., 2007*) and rate of cocaine delivery to the central nervous system affects the magnitude of behavioural responses (*Samaha & Robinson, 2005*), we decided to test if the number of foraging trips was affected following the more rapid volatilised treatment method (*Søvik, Cornish & Barron, 2013*). In a flight cage bees were trained to visit an *ad libitum* sucrose feeder where they were given individually distinctive paint marks. Bees that returned five times after being marked were caught and treated with 5 μg volatilised freebase cocaine or control. We chose 5 μg as this was the highest volatilised dose previously tested that did cause deleterious motor effects (*Søvik, 2013*). Bees were assigned to treatment groups randomly. The number of visits treated bees made to the feeder in the 40 min following treatment were recorded. Sucrose concentration has previously been shown to affect foraging rate in bees (*Seeley, 1995*), studied responses of bees to both low (0.5 M) and high (2.0 M) sucrose solutions.

## Effects of volatilised cocaine on sucrose responsiveness

To test if volatilised cocaine affected sucrose responsiveness we used cage-reared bees of known age and social history. Upon emergence, bees were placed in mesh cages ($20 \times 16 \times 3$ cm) with *ad libitum* access to honey. The cages contained eighty bees each and were kept at 34 °C for 6 days. When the bees were 7 days old, they were fastened individually in an 8 mm tube in a way that prevented the bees from escaping but allowed the proboscis and antenna to move freely (*Bitterman et al., 1983*). This method is most commonly used for proboscis extension learning experiments (*Felsenberg et al., 2011*) but has also been used to measure bees' responsiveness to sucrose (*Scheiner, Page Jr & Erber, 2004*). Once harnessed,
bees were treated with 0 or 10 µg volatilised cocaine and tested for sucrose responsiveness. The 10 µg was chosen based on an initial pilot experiment suggesting that this dose was sufficient to elicit increased responsiveness to sucrose (E Søvik, 2012, unpublished data). We repeated this experiment with 0, 5, 10, 20, or 50 µg volatilised cocaine to examine if the effect seen with 10 µg was dependent on the cocaine dose used.

The sucrose responsiveness test consisted of touching a drop of 10% sucrose solution to the antennae of bees 3 min after drug exposure, and recording whether or not the proboscis was extended. After the test, bees were tested for their response to water and honey. Bees responding to water, or failing to respond to honey were excluded from the analysis.

### Effects of volatilised cocaine on responsiveness to electric shocks

To examine effects of cocaine on responsiveness to electric shock, bees were fastened between two conducting brass plates with a piece of electrical tape (for details see *Vergoz et al., 2007*). After treatment with 0, 5, 10, 20 or 50 µg volatilised cocaine, brass plates were connected to an electrical supply, and bees were shocked with gradually increasing voltage (0.5 V every 5 min) from 0 to 10 V. The first voltage at which a bee extended its stinger (a reflexive response) was recorded for each bee. Testing occurred in front of an extraction fan so no alarm pheromone would linger in the testing room and affect bees yet to be tested (*Vergoz et al., 2007*). Comparisons between groups were based on $EV_{50}$ (half maximal effective voltage): the point at which half of all bees in the treatment group extended their stingers.

## RESULTS

### Effects of cocaine on honey bee foraging preferences

Repeatedly treating bees with 3 µg cocaine in DMF at a sucrose feeder enhanced bees' preference for that feeder in a choice assay when compared to bees treated with DMF as a control (Mann–Witney test: $U = 2,185$, $p = 0.0038$; Effect size: $r = -0.25$). Treating bees with cocaine at a feeder while they were foraging resulted in a greater preference for that feeder in a free-choice test when compared to bees treated with DMF (vehicle control) while foraging at the feeder (Fig. 2A).

### Effect of volatilised cocaine on foraging rate

Bees treated with 5 µg volatilised cocaine once at a 0.5 M feeder made significantly more return visits to the feeder in the 40 min following treatment, than controls ($t_{70} = 5.0710$, $p = 0.00003$; Effect size: $d = 0.9905$; Fig. 2B). Bees treated with cocaine at a 2 M feeder showed no increase in visitations after cocaine treatment ($t_{70} = -0.2087$, $p = 0.8353$; Effect size: $d = 0.0399$; Fig. 2B). This demonstrated that bees altered the rate at which they returned to a low quality feeder following volatilised cocaine treatment, but not to a high quality feeder (Fig. 2B).

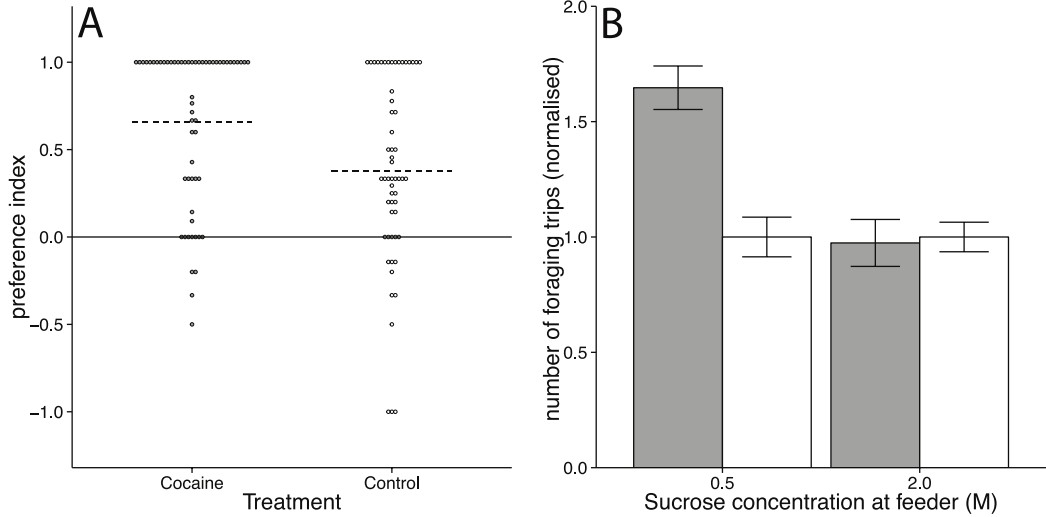

**Figure 2 Foraging behaviour in honey bees following cocaine administration.** (A) Scatter plot showing the effect of topical cocaine treatment on preference for the green arm. Each point represents one bee. Dotted lines mark median values for each treatment group. The preference for the green arm was significantly higher for cocaine-treated than control-treated bees (Mann–Witney $U = 2,185, p = 0.0038$). (B) Effect of volatilised cocaine treatment on visitation rate at a sucrose feeder (error bars represent standard error). Bees treated with volatilised cocaine (grey bars) increased their rate of foraging relative to controls (white bars) when foraging at a 0.5 M sucrose feeder ($t_{70} = 5.0710, p = 0.00003$), but not at a 2 M sucrose feeder ($t_{70} = -0.2087, p = 0.8353$).

## Effects of volatilised cocaine on sucrose responsiveness

Treatment with 10 µg of volatilised cocaine increased bees responsiveness to sucrose ($\chi^2 = 6.0268$, $df = 1$, $p = 0.0141$; Effect size: $d = 0.6331$; Fig. 3A). The effect was dependent on the cocaine dose. Bees treated with 5 and 10 µg of cocaine were significantly more responsive to sucrose than controls ($\chi^2 = 14.089$, $df = 4$, $p = 0.0070$; Fig. 3B), while bees treated with 20 or 50 µg of cocaine did not differ from controls. The control treatment differed quite markedly between two experiments; however, this is likely because the two experiments were performed at different times of the year. Sucrose responsiveness varies with season and environmental conditions. The important aspect is the difference between the cocaine treated bees and the control treated bees in a given experiment.

## Effects of volatilised cocaine on responsiveness to electric shocks

Cocaine affected bees' responsiveness to shock in a dose dependent manner (Fig. 3C). We used the $EV_{50}$ for statistical comparisons. All bees treated with cocaine were significantly more sensitive to electric shock than control treated bees ($F_{4,40} = 5.4$, $p = 0.0015$; Fig. 3C). There were no differences between the cocaine treatment groups with the exception of bees treated with 50 µg cocaine. The bees treated with 50 µg were significantly more sensitive than all other cocaine treated groups. The $EV_{50}$ of cocaine treated bees (50 µM $EV_{50} = 2.1$; 20 µM $EV_{50} = 3.5$; 10 µM, $EV_{50} = 2.6$; 5 µM $EV_{50} = 3.1$) was lower than in control treated bees ($EV_{50} = 5.3$).

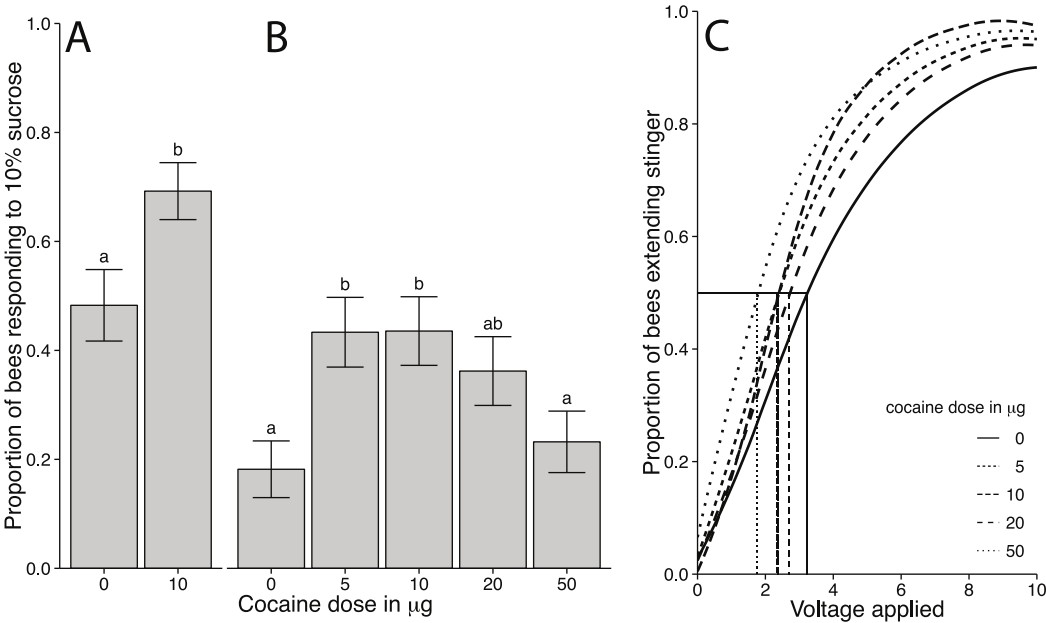

**Figure 3 Behavioural responsiveness following cocaine administration in honey bees.** (A) Proportion of bees responding to 10% sucrose following treatment with 0 or 10 μg of volatilised cocaine (error bars represents standard error and letters denote statistically different groups). There was a significant increase in sucrose responsiveness in bees treated with 10 μg cocaine relative to control ($\chi^2 = 6.1013$, $df = 1$, $p = 0.0135$). (B) Proportion of bees responding to 10% sucrose following treatment with 0, 5, 10, 20, or 50 μg of volatilised cocaine. There was a dose-dependent relationship between cocaine dose and sucrose responsiveness ($\chi^2 = 14.089$, $df = 4$, $p = 0.0070$). (C) Shock responsiveness of bees following cocaine administration. Curves are based on weibull distributions of shock responsiveness for each group. Comparisons are based on estimates of $EV_{50}$ for 40 bees per group ($F_{4,40} = 5.4$, $p = 0.0015$). Pairwise comparisons found that the 50 μg group was different from all other groups, while the remaining cocaine treated groups were different from controls.

## DISCUSSION

In two separate experiments we observed that cocaine administration affected aspects of foraging decisions. Cocaine treatment increased the preference for a feeding location, and the rate of visitation at a sucrose feeder (Fig. 2). Further, cocaine caused increased responsiveness to sucrose (Figs. 3A and 3B). These findings, as well as those of *Barron et al. (2009)*, lends support to the hypothesis that cocaine alter reward responses across divergent animal groups. However, we also found that cocaine made bees more responsive to electric shock (Fig. 3C). Thus, the effect of cocaine is not limited to reward-related behaviours. Rather cocaine altered a range of behavioural responses, all, at least partially, modulated by octopaminergic or dopaminergic signalling. This is consistent with cocaine broadly interfering with octopaminergic and/or dopaminergic signalling in honey bees.

Our experiments indicate that cocaine alters the perceived concentration of sucrose in honey bees. Previous studies have shown that bees form stronger associations when rewarded with higher sucrose concentrations compared to lower ones (*Loo & Bitterman, 1992*). This can potentially explain the increased response rate to 10% sucrose. Interestingly, cocaine only caused bees to increase their visitation rate at the low sucrose

concentration feeder. This could be because at high sucrose concentrations, the relative change in perceived sucrose concentration is lower than with low sucrose concentrations.

This study provides further support to the bold claim that the neurochemicals modulating reward systems are broadly conserved across diverse animal phyla (*Barron, Søvik & Cornish, 2010*; *Waddell, 2013*), and therefore despite certain differences in specific neurochemistry and transporter affinities, diverse reward systems appear susceptible to disruption by the same drugs (*Søvik & Barron, 2013*). By 'broad conservation' we do not imply that the reward processing circuitry present in insects and mammals was present in the last common ancestor of these groups, but rather that biogenic amines may have performed functions in the common ancestor that predisposed them to become modulators of reward systems in most animal phyla (*Barron, Søvik & Cornish, 2010*).

We believe that this is not necessarily contradictory to the ecological function of cocaine as a deterrent compound inhibiting herbivory of the coca plant. Cocaine also enhanced responsiveness to electric shock (Fig. 3C), and our previous work has shown cocaine profoundly damaged motor systems, coordination and locomotion in bees (*Søvik, Cornish & Barron, 2013*). Similar findings have been reported for other insects, emphasising the insecticidal properties of cocaine (*Nathanson et al., 1993*). The effects of cocaine on insects are therefore extremely dose dependent. The rewarding effects reported here were seen at very low doses only. When herbivores ingest plant tissues containing cocaine, they quickly ingest enough to interfere with their motor system, and thus cannot continue feeding (*Nathanson et al., 1993*).

In mammals it is also seen that in recreational drug use, drugs are usually administered in ways that bypass the gut and achieve rapid delivery of a very low and controlled dose to the central nervous system in order to maximise the hedonic effects while minimising the toxic effects (*Hagen et al., 2009*).

Given the similarities observed in drug responses between vertebrate and invertebrates, it might be possible to use simple invertebrate animals as models for studying aspects of drug reward. While much important work is being done with mammalian models, many other fields of neuroscience have benefitted greatly from the advantages of relatively simple invertebrate model systems (*Burne et al., 2011*). Previous work with *Drosophila* has highlighted the importance of circadian regulation (*Andretic, Chaney & Hirsh, 1999*; *Abarca, Albrecht & Spanagel, 2002*) and LIM-only proteins (*Heberlein et al., 2009*; *Lasek et al., 2010*) for the formation of sensitisation. However, invertebrate research has so far not been particularly concerned with drug reward (*Søvik & Barron, 2013*). Given the importance of drug reward in human drug use (*Siegel, 2005*), this should be a key area for future investigations. Honey bees spend the majority of their time searching out natural rewards in their environments and have a long history as a model organism for studying the neurobiology of natural rewards (*Perry & Barron, 2013*). Considering the similarities in responses to cocaine between humans and bees, we can now capitalise on the potential of the honey bee as a simple invertebrate model organism to study drug reward.

## ACKNOWLEDGEMENTS

We would like to thank Falk von Hollen for helping with the foraging preference experiments, and Lea Denneulin and Flavie Rongère for helping with the shock responsiveness experiment.

### Funding

This work was supported by the Australian Research Council Grant DP0986021 awarded to ABB. The funders had no role in study design, data collection and analysis, decision to publish, or preparation of the manuscript.

### Grant Disclosures

The following grant information was disclosed by the authors:
Australian Research Council: DP0986021.

### Competing Interests

The authors declare there are no competing interests.

### Author Contributions

- Eirik Søvik conceived and designed the experiments, performed the experiments, analyzed the data, wrote the paper, prepared figures, reviewed drafts of the paper.
- Naïla Even performed the experiments, analyzed the data, reviewed drafts of the paper.
- Catherine W. Radford performed the experiments, reviewed drafts of the paper.
- Andrew B. Barron conceived and designed the experiments, contributed reagents/materials/analysis tools, wrote the paper, reviewed drafts of the paper.

### Supplemental Information

Supplemental information for this article can be found online at http://dx.doi.org/10.7717/peerj.662#supplemental-information.

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
