# Peer review of "Cocaine affects foraging behaviour and biogenic amine modulated behavioural reflexes in honey bees"

_PeerJ, doi:10.7717/peerj.662_

## Round 0.1 · original submission · Major Revisions

This report deals with a study on behavior of honey bees during effects of topical and volatilized cocaine administration in three assays. We see the importance of this study. However, improvements suggested by the reviewers are important to address as this will improve overall quality of this study. Hence, I recommend you to improvise this MS and also deposit point to point rebuttal to the questions raised by the reviewers.

·

Basic reporting

This article reports a study where effects of topical and volatilized cocaine administration on behavior of honey bees in three assays have been investigated. The basic reporting is incomplete for coverage of relevant prior literature. There also are a few writing mistakes.

Background and prior literature:
The first behavioral assay is for foraging preference to feeders placed in a green and white vs blue and white tunnel. Topical cocaine administration alters preference rate for the green-white tunnel more than control treatment. However, there is no discussion of why green vs blue were compared, and pertinent literature on bee color vision is not included. The described preference and preference index are used previously in honey bees and elsewhere but authors do not mention the relevant literature (see Agarwal et al. PLoS ONE 2011). Incidentally, the Agarwal and colleagues discuss, probably for the first time, and provide evidence that it is too simplistic to see dopamine equals punishment and octopamine equals reward in honey bees.

The authors also show that foraging rates or visitation rates to feeders are altered for the lower sucrose concentration feeder (0.5M) when bees are administered volatilized cocaine and not for the higher sucrose concentration (2M).

Sucrose responsiveness and sting extension response are the two other measures examined in response to cocaine administration. Not withholding many omissions, literature on sucrose responsiveness is more complete. In sting extension response, unfortunately, relevant literature is not included and comparisons are not made. For instance the claim by authors that cocaine targets different biogenic amines has support only if role of serotonin in sting extension response were to be discussed (see Tedjakumala et al. 2013). Similarly, the described sting extension response threshold change (change in EV50, as per authors) has been described for another "dirty drug" recently (Giannoni-Guzman et al. 2014 ethanol effect on sting extension response).

The authors cite the Vergoz et al. 2007 to support the argument that sting extension response is "aversive". However, the group that published the Vergoz et al. 2007 paper, later examined if indeed sting extension was predictive of aversive behavior. This distinction, and the supporting literature should be cited. For instance, sting extension could be an aggressive response instead.

Writing issues:
"perception of" probably should be changed to "response to" when reporting on honey bee response to sucrose concentration.

"Finally" at the end of introduction when describing the experiments gives a sense of exhaustion. Perhaps, "lastly" could be used.

"forging" where it appears should be "foraging"

Experimental design

Experimental design appears appropriate but without sufficient detail or supporting information it is difficult to judge if all claims are supported by the presented data.

Specifically, the authors should address the following:
The use of tunnels with different colored entrances was interesting, however it is not discussed how this method compares to say visit to artificial flowers with different colors. (of course again the pertinent literature, starting with von Frisch).

In administration of cocaine, were the tissue titres/amounts reached examined with the employed treatment methods? Time course of these titres would also be important. There is experimental evidence cited by the authors for the topical application (the effect lasts 1.5 hrs). How does this effect correlate with time course of cocaine in hemolymph, in brain?? How about the volatilized cocaine method? The foraging rates were examined for 40 minutes. Do we know how long the effect or the chemical lasts?

In the sting extension response study a continuously increased voltage application was used but rate of increase in voltage is not reported (1mV per minute or 1V per minute???). It would be difficult to compare to other studies where voltage was increased either at a constant rate or in steps at constant periods if this rate is not known. The Giannoni-Guzman et al. 2014 publication describes a change in the response curve, and that the first sting response to a voltage level is not predicting consistent response in presence of a drug. Are the reported sting response voltages in this study represent the threshold beyond which the bees always responded by sting extension or the first response may be an erratic response due to drug effects?

Incidentally, the authors have compared the EV50 or shock voltage level when 50% of individuals responded by sting extension across the treatment groups. A visual aid for this comparison is needed on the plot for the sting extension reponse cumulative curves. A horizontal line that cuts across the response curves at the 50% level would be useful.

Validity of the findings

The validity of the findings depend on clarity of experimental design. Explanations or responses to the above points in experimental design would be needed before this could be fully decided.

Reviewer 2 ·

Basic reporting

Pass

Experimental design

The experimental design seems to be all right, but some more description and justification is needed.

Validity of the findings

Some more data related to the "training of the bees" are required. Additional data in support of "shock response" is needed.

Additional comments

The manuscript submitted by Sovik et al. in PeerJ is interesting and novel. The data provided is interesting. However, there are several issues that need to be addressed before it can be finally accepted. Also it needs additional experiments/additional data.

General comments:

In the introduction part, the concept of biogenic amines produced in plants and their effects on animals (insects to mammals) is weekly written. This portion need to be justified with more examples, exceptions and refs. It will be much better if the authors can comment on such concept at the discussion section again based on their findings and debate on that.

The more specific questions:
The tubes used in this study, i.e. to approach for foraging are not same (one with horizontal line and with green strips while the other one is vertical blue lines). A representative image of the experimental system should be added in the manuscript for better understanding. Reference for such system should be added. I feel that the entire data regarding the training course should be a part of the manuscript. The “unpublished” data (Page 6, line 113) should be available for proper reviewing.

The authors mentions that “the green tunnel was reinforced more than the blue” is expected. Why is such, that need to be clarified in more details in the light of the classic reference (Srinivasan et al. 2000 Science) paper.

Page 7, line 125: Only 5 individuals were used for this experiment. I feel that the number can be increased further for a better reliable data.

Figure 2: In part A and part B, the parameters in the Y axis is same (Proportion (and standard error) of bees responding to 10 % sucrose). Still the values at the 0 and 10 μg of volatilized cocaine are different in A and B. The authors need to comment on that. I failed to understand why there are differences in this two cases.

Figure 2C is poorly represented. The effect of cocaine on Shock responsiveness is not well convincing. More in-depth analysis and more experiments in order to characterize that part is required.

Important:

As mentioned and demonstrated that cocaine has some effect on the dance pattern/or forging decisions, it is important to check or even comment on if cocaine can alter other biological functions and/or other decision making abilities in bees. How this decision making abilities are different than general toxicity effects? This is an important question to address. The authors need to show that cocaine does not have significant toxicity (in the doses used in this study) on the bees. Or if it has any toxicity? The authors need to comment on that.

The manuscript should compare the effect of cocaine and other similar compounds (in the light of CART peptide and corresponding receptors) present in the mammal and insects. Can authors comment on that aspects using honey bee as a model system? That will increase the weight of the paper. I could not find a solid discussion on that part, particularly the neuronal transmission system (including the reward system) which can be modulated by cocaine in bees. A comparative model in the discussion section can be a very good addition. This model should compare bees with mammals as that was the starting point of this MS.

·

Basic reporting

The study is well conducted and claims appear to be well substantiated based on the information presented in the paper.

The figure legend describes the figures well but small changes in language can make the legend describe the finding and be more effective. I would recommend revisiting figure legends.

Additionally, in the methods as well as figure legends please report quantity in terms of molar concentrations instead of µg/ml .

Experimental design

The experimental design is sound. I have a suggestion, which is not a must for acceptance of paper but can add value to the study. If the authors have access to GC or any other measurement approach, I would like authors to study the internal cocaine levels in bees as a function of time. It would be nice to see the internal amount and rewarding behavior, as well desensitization to cocaine if any over duration of one trial.

Given that topical vs. volatilized cocaine treatment is important methodological detail, it would be nice if authors can present that data in this paper.

Validity of the findings

From the information presented the claims are well backed.

Additional comments

I like the introduction on herbivory and selection of compounds to deter insects as a means to set up the question.

I like the cautious wording in discussion such as “By ‘broad conservation’ we do not imply that the reward processing circuitry present in insects and mammals was present in the last common ancestor of these groups, but rather that biogenic amines may have performed functions in the common ancestor that predisposed them to become modulators of reward systems in most animal phyla (Barron et al., 2010)”.

---

## Round 0.2 · Minor Revisions

Please take care of the final comments of reviewer#1, before I come to final decision.

·

Basic reporting

The submission is well prepared, and all previous inquiries have been addressed adequately. The only exception is the pharmacokinetics of cocaine in the honey bee. The authors may explicitly state these measurements were not done, and indicate why they think this would or would not effect their conclusions. In other organisms the cocaine half-life is about 20 minutes, although active metabolites may remain longer in the tissues. Some discussion of this caveat may help future directions and also may help in interpretation of current manuscript.

Experimental design

Experimental design is now clearly explained, and it is appropriate for the questions addressed.

Validity of the findings

The data are presented fully and statistics are appropriate. The conclusions are stated appropriately. The concerns were fully addressed.

Additional comments

Please consider adding a few sentences addressing the cocaine pharmacokinetics.

---

## Round 0.3 · accepted · Accept

Congratulations!!!, This manuscript is accepted.